# Structural mechanisms of pump assembly and drug transport in the AcrAB–TolC efflux system

**Xiaofei Ge[1,2]\*, Zhiwei Gu[1], Jiawei Wang[1]\***

[1]State Key Laboratory of Membrane Biology, Beijing Frontier Research Center for Biological Structure, School of Life Sciences, Tsinghua University, Beijing, China; [2]Health and Wellness, City University of Macau, Macau, China

## eLife Assessment

Ge et al here report a structural study of the native tripartite multidrug efflux pump complexes from *Escherichia coli* that identifies a novel accessory subunit, YbjP, the structure of the native TolC-YbjP-AcrABZ complex, as well as structures of the AcrB protein in L, T, and O conformations. The strength of the structural data is **compelling**, and the importance of the findings is potentially **fundamental**. In the revised manuscript, the authors have included additional analysis and made comparisons with pre-existing data which has helped place the data and its impact in the proper context.

**\*For correspondence:**
gxf16@tsinghua.org.cn (XG);
jwwang@tsinghua.edu.cn (JW)

**Abstract** Tripartite multidrug efflux pumps that span the cell envelope are essential for antibiotic resistance in Gram-negative bacteria. Here, we report cryo-EM structures of two endogenous efflux complexes from *Escherichia coli*: a TolC–YbjP subcomplex at 3.56 Å resolution and the complete TolC–YbjP–AcrABZ pump at 3.39 Å. Structural analysis reveals that YbjP, a previously uncharacterized lipoprotein, binds TolC in a 3:3 stoichiometry, bridging the TolC protomers at their equatorial domain. Clear density of the mature YbjP's N-terminal Cys19 indicates that YbjP is anchored to the outer membrane by an N-terminal lipid moiety. Notably, YbjP remains bound as TolC undergoes AcrA-induced opening, suggesting that this accessory protein accommodates the conformational change. The AcrB trimer simultaneously presents three distinct conformational states (L, T, and O), capturing a complete transport cycle. These high-resolution structures provide insights into the architecture and mechanism of clinically relevant efflux machinery, identifying YbjP as a previously unrecognized structural component that contributes to TolC positioning, and may assist in its membrane localization.

## Introduction

Tripartite efflux pumps are essential for Gram-negative bacteria to expel a wide range of toxic compounds, including antibiotics, across their dual-membrane cell envelope (*Poole, 2005*; *Li et al., 2015*; *Du et al., 2015*; *Du et al., 2018*; *Blair et al., 2014*). Among these systems, the AcrAB–TolC complex of *Escherichia coli* is one of the most thoroughly studied (*Li et al., 2015*; *Wright et al., 2025*). It comprises the inner membrane RND (Resistance–Nodulation–Division) transporter AcrB, the periplasmic membrane fusion protein AcrA, and the outer membrane channel TolC (*Du et al., 2018*; *Wright et al., 2025*; *Ma et al., 1995*; *Nikaido, 2009*; *Okusu et al., 1996*). AcrA, anchored to the inner membrane via N-terminal lipidation, bridges AcrB and TolC, mediating conformational signaling essential for TolC gating (*Koronakis et al., 2000*; *Mikolosko et al., 2006*; *Murakami et al., 2002*). The assembling pathway of the tripartite pump involves the formation of an AcrA–AcrB subcomplex,

followed by the recruitment of TolC to form the complete efflux apparatus (*Shi et al., 2019*). In its isolated form, the periplasmic tip of TolC adopts a closed conformation, stabilized by interactions between the α-helices (*Pei et al., 2011*; *Andersen et al., 2002a*; *Bavro et al., 2008*). Structural studies have shown that TolC transitions from a closed to an open state in response to AcrAB assembly, a process crucial for efflux activation (*Shi et al., 2019*; *Wang et al., 2017*).

Despite significant progress in structurally characterizing the AcrAB–TolC system, fundamental questions remain regarding how TolC is positioned and retained in the outer membrane. Unlike homologous channels such as *Pseudomonas aeruginosa* OprM (*Phan et al., 2010*) and *E. coli* CusC (*Kulathila et al., 2011*), which possess covalently attached N-terminal lipids, TolC lacks a lipid anchor. This raises the possibility that *E. coli* may utilize accessory factors to localize and stabilize TolC during pump assembly. Furthermore, while prior studies using crystallography and cryo-EM have revealed low-to-medium resolution snapshots of the assembled pump (*Koronakis et al., 2000*; *Mikolosko et al., 2006*; *Murakami et al., 2002*; *Shi et al., 2019*; *Wang et al., 2017*; *Eicher et al., 2012*), the limited resolution and conformational heterogeneity in those structures impeded accurate modeling of flexible regions and may have masked the presence of additional subunits.

Here, we address these knowledge gaps by determining two high-resolution cryo-EM structures of the TolC-based efflux machinery in *E. coli*. The first is a 3.56-Å structure of a TolC–YbjP subcomplex, and the second is a 3.39-Å structure of the fully assembled TolC–YbjP–AcrABZ pump. These structures reveal a previously uncharacterized outer membrane lipoprotein, YbjP, bound to TolC, and shed light on how TolC is secured and transitions to the open state. Together, these findings enhance our understanding of the architecture, assembly, and regulation of this clinically important efflux pump.

## Results

### Structure of a TolC–YbjP closed-state complex

Endogenous purification of *E. coli* membranes yielded a stable assembly containing the outer membrane channel TolC (*Figure 1—figure supplement 1A*). The complex was purified directly from *E. coli* as a stoichiometric assembly, without the need for artificial fusion or crosslinking. Cryo-EM analysis resolved the complex at 3.56-Å resolution (*Figure 1—figure supplement 1B–E*, *Supplementary file 1*). TolC forms a homotrimer composed of a β-barrel that spans the outer membrane and an elongated periplasmic α-helical domain (*Figure 1A–D*). The periplasmic α-helical domain comprises an α-helical barrel extended by two helix-turn-helix (HTH) motifs at its periplasmic extremity (TolC repeat 1: helices H3 and H4; repeat 2: helices H7 and H8). These structural elements converge at an equatorial domain (*Figure 1C, D*). The inward-bending HTH coiled-coils occlude the periplasmic tunnel, maintaining TolC in a closed conformation – consistent with previous crystal structures (*Koronakis et al., 2000*; *Figure 1C*).

An unexpected density at the periplasmic face of TolC indicated the presence of an additional component (wheat in *Figure 1A, B*). An initial backbone trace, followed by a DALI search (*Holm et al., 2023*) tentatively matched Tai3, a periplasmic type IV immunity protein associated with the T6SS amidase effector Tae3 (PDB ID: 4HZ9) (*Dong et al., 2013*) from *Ralstonia pickettii*. However, discrepancies in side-chain density and species origin indicated that the match was likely incorrect. Systematic screening of the AlphaFold Database (*Varadi et al., 2022*) using CryoNet (*Xu, 2019*) identified *E. coli* YbjP (UniProt P75818) as the top candidate, with all side chains matching the experimental density (*Figure 1—figure supplement 2*). YbjP features a globular domain and a structured N-terminal loop stabilized by a Cys36–Cys144 disulfide bond (*Figure 1E, F*). The N-terminal loop of YbjP adopts an extended conformation parallel to the Helix 2 (residues 38–61) of TolC's α-helical barrel, forming substantial intermolecular interfaces. Buried surface area analysis reveals 787 Å² of contact with the primary TolC protomer (green in *Figure 1F*), and 1037.4 Å² with the adjacent protomer (light green), suggesting asymmetric binding energetics. The N-terminal density extends to Cys19, which corresponds to the predicted signal peptide cleavage site, positioning the lipoprotein at the periplasmic face of the outer membrane.

UniProt predictions indicate that YbjP is dually lipidated via *N*-palmitoyl and *S*-diacylglycerol modifications, allowing its anchorage to the outer membrane. This represents an elegant evolutionary solution, as while many Gram-negative TolC homologs (e.g., *P. aeruginosa* OprM, *Phan et al., 2010*; *E. coli* CusC, *Kulathila et al., 2011*; etc.) possess native lipid anchors (*Figure 1—figure supplement*

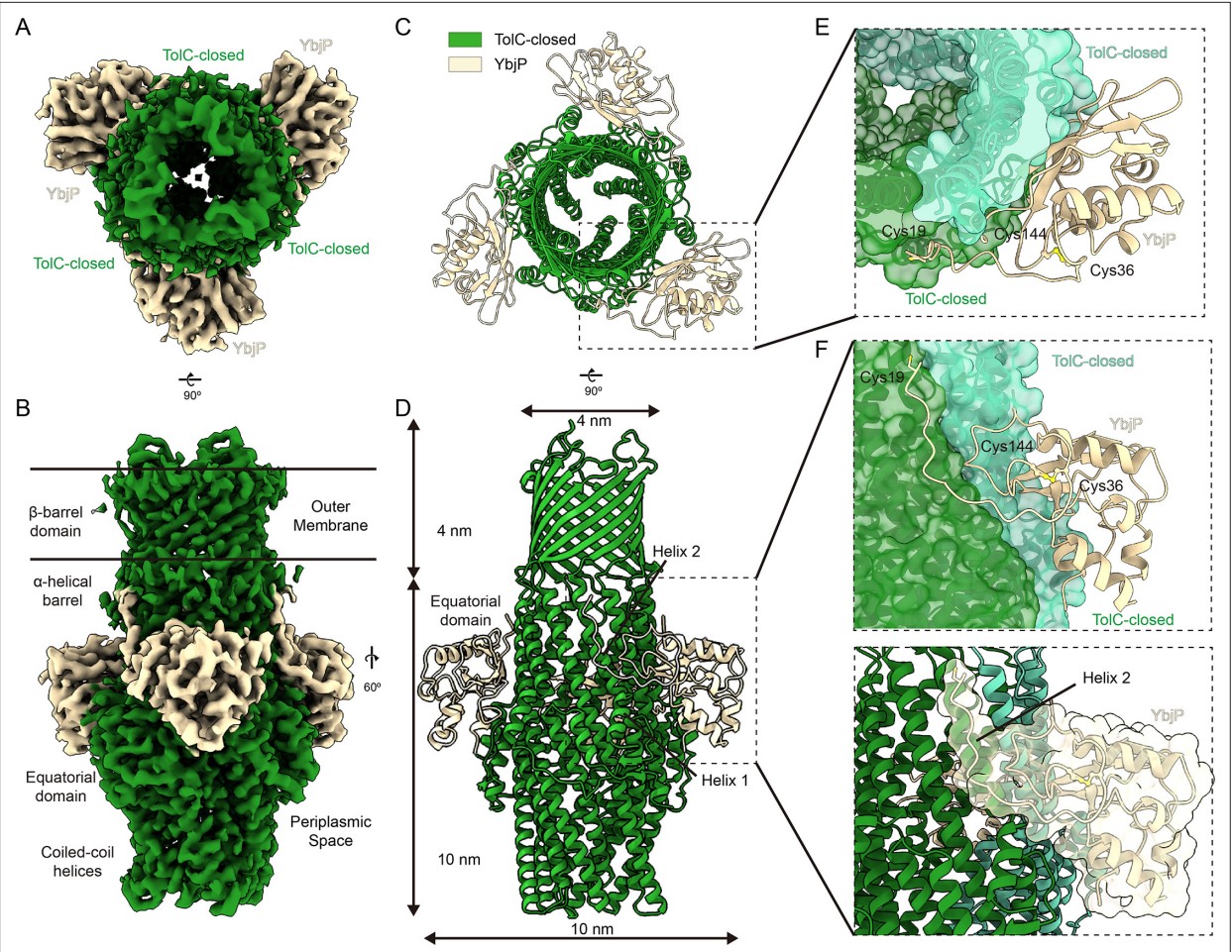

**Figure 1.** Structure of the TolC–YbjP complex (closed state). (**A, B**) Cryo-EM density map of the endogenous TolC–YbjP complex at 3.56 Å resolution, shown in top and side views, spanning the outer membrane (OM) and periplasmic space. TolC forms a homotrimeric 'channel-tunnel' ~14 nm long, with its 12-stranded β-barrel embedded in the OM and periplasmic coiled-coils sealed at the tip. YbjP (wheat) wraps around the equatorial domain of TolC, with continuous density linking YbjP's N-terminal Cys19 into the OM micelle region. (**C, D**) Top view (from the outer membrane) and side view of the TolC–YbjP cartoon model, showing three YbjP molecules arranged symmetrically around TolC's threefold axis. (**E, F**) Each YbjP straddles the interface between two TolC protomers, occupying a groove on the TolC surface. TolC is shown as surface and cartoon (each protomer in a different green color) and YbjP in wheat cartoon. YbjP's globular domain inserts between TolC protomers, while an N-terminal linker connects to the OM.

The online version of this article includes the following figure supplement(s) for figure 1:

**Figure supplement 1.** Cryo-EM processing workflow for TolC–YbjP complex.

**Figure supplement 2.** Local EM density of YbjP in TolC–YbjP complex.

**Figure supplement 3.** Structural comparison of the anchoring domains from six prototypical efflux pump outer membrane factors.

---

*3*), *E. coli* TolC instead recruits YbjP as a dedicated membrane-tethering partner. The YbjP globular domain nestles between adjacent TolC equational domains, forming a stable 3:3 complex (*Figure 1B*). The identification of YbjP, along with its defined membrane anchoring and structural compatibility with TolC, suggests a new class of outer membrane partners that may regulate the TolC-dependent pathways. Notably, the structure of TolC shows no conformational changes upon YbjP binding when compared to the free, closed form of TolC (*Koronakis et al., 2000*).

## Structure of TolC–YbjP–AcrABZ complex

An improved 3.39 Å resolution structure of the endogenous AcrAB–TolC complex from *E. coli* was determined, revealing well-resolved density across the entire assembly (*Figure 2—figure supplement 1*, *Supplementary file 1*). This structure allows for the accurate modeling of side-chain interactions throughout the tripartite channel, surpassing earlier models in completeness and resolution

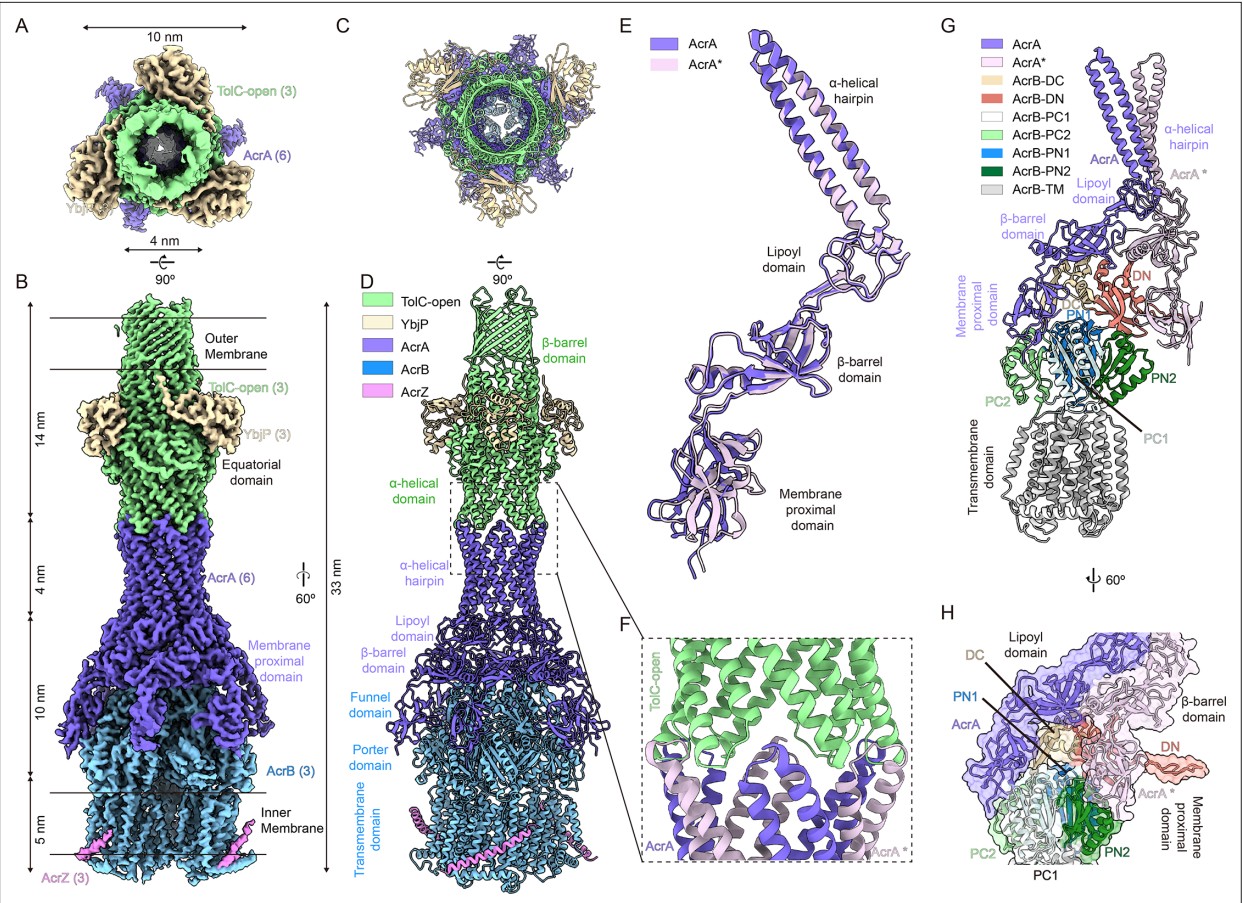

**Figure 2.** Architecture of the fully assembled TolC–YbjP–AcrABZ efflux pump. (**A, B**) Top and side views of the cryo-EM density map (3.39 Å) of the endogenous TolC–YbjP–AcrABZ complex, with the inner membrane (IM) and outer membrane (OM) boundaries indicated (black bars). The map reveals the complete pump spanning 33 nm. Densities for TolC (green) are supposed to be in the OM region, AcrB (blue) in the IM region as a trimer, and six AcrA protomers (purple) forming an elongated barrel bridging TolC and AcrB. Three AcrZ protomers (pink) are seen at the periphery of the AcrB trimer in the IM region. (**C, D**) Cartoon models of TolC–YbjP–AcrABZ in top and side views. Three AcrB protomers (blue) form a trimeric base, surrounded by six AcrA molecules (purple) and three AcrZ helices (pink). Each AcrB protomer consists of funnel, porter, and transmembrane domains. (**E**) Structural alignment of the two AcrA conformations (AcrA and AcrA*) reveals differences in the orientation of the membrane-proximal domain. (**F**) Zoomed-in view of the TolC–AcrA interface, showing the tight interaction between TolC's periplasmic helix-turn-helix motifs and AcrA's α-helical hairpin domain. (**G, H**) Close-up views of the interface between AcrA (purple), AcrA* (light purple), and AcrB. Panel (**G**) shows the frontal cartoon representation; panel (**H**) displays a 60° rotated view, highlighting the intimate packing of AcrA with AcrB.

The online version of this article includes the following figure supplement(s) for figure 2:

**Figure supplement 1.** Cryo-EM processing workflow for TolC–YbjP–AcrABZ complex.

**Figure supplement 2.** Local EM density of TolC and AcrA in TolC–YbjP–AcrABZ complex.

**Figure supplement 3.** Local EM density of AcrB and AcrZ in TolC–YbjP–AcrABZ complex.

(*Wang et al., 2017*; *Figure 2—figure supplements 2 and 3*). YbjP remains bound to TolC (*Horne et al., 2025*), suggesting that this lipoprotein is stably retained during pump assembly and functions as a molecular clamp that anchors TolC to the outer membrane during AcrA and AcrB engagement (*Figure 2A–D*).

The TolC–YbjP–AcrABZ complex adopts a funnel-like architecture spanning the cell envelope, consistent with prior models of tripartite pumps (*Wang et al., 2017*; *Figure 2B*). TolC, tethered to YbjP, caps the outer membrane end and docks onto a hexameric ring of AcrA adaptors in the periplasm, which in turn surrounds the AcrBZ trimer embedded in the inner membrane (*Figure 2B, D*). The full assembly is about 33 nm tall, comprising a 14-nm TolC channel, a 14-nm periplasmic portion of AcrAB, and a 5-nm transmembrane portion of AcrBZ, matching the distance between the two membranes (*Figure 2B*).

Cryo-EM density reveals six AcrA molecules positioned beneath each TolC trimer (*Figure 2D*), yielding a 3:6:3 stoichiometry for TolC:AcrA:AcrB. AcrA therefore forms an elongated hexamer that bridges TolC and AcrB. This tripartite architecture is stabilized by three distinct sets of interfaces: (1) contacts between the AcrB trimer and the basal regions of AcrA, (2) extensive AcrA–AcrA lateral interactions within the hexameric ring, and (3) tip-to-tip junctions formed between the upper AcrA α-helical hairpin and the periplasmic entrance of TolC (*Figure 2D*). While each AcrA protomer maintains the characteristic four-domain architecture – comprising the α-helical hairpin, lipoyl, β-barrel, and membrane-proximal (MP) domains – functional asymmetry is observed in their interactions (*Figure 2E*). The trimeric TolC, which contains six HTH motifs due to internal repeats (TolC repeat 1/2), engages the AcrA hexamer via quasi-equivalent binding: adjacent AcrA and AcrA* protomers interact differentially with the intra- and inter-protomer grooves of TolC, respectively (*Figure 2F*). The primary structural difference between AcrA and AcrA* lies in the configurations of their MP domains (*Figure 2E*).

AcrB assembles as a homotrimer on the inner membrane, with each protomer comprising three distinct domains: a funnel/docking domain, a porter/periplasmic domain, and a transmembrane domain (*Figure 2D*). The funnel domain consists of two subdomains, DN and DC (denoting the N- and C-terminal subdomains) (*Figure 2G*). The porter domain is organized into four subdomains – PN1, PN2, PC1, and PC2 – named for the N- and C-terminal subdomains. When viewed from the periplasmic side, these subdomains are arranged in a clockwise orientation, with PN1 positioned closest to the central axis of the trimer. In the hexameric AcrA assembly, the lipoyl and β-barrel domains form two stacked concentric rings. While the lipoyl domains remain uninvolved in AcrB binding, the β-barrel domains specifically engage the funnel domain of AcrB. Strikingly, the MP subdomains of AcrA exhibit asymmetric binding: AcrA–MP interacts with the DC and PC1 subdomains of AcrB, whereas AcrA*–MP contacts PN1 and DN via an extended loop, respectively (*Figure 2G, H*). This differential engagement of MP subdomains establishes the structural basis for the observed structural divergence between AcrA and AcrA* (*Figure 2E*).

Our density maps clearly resolve the small transmembrane protein AcrZ (49 amino acids) bound to each AcrB protomer (*Figure 2B, D*). AcrZ adopts a helical structure that is embedded within a hydrophobic groove formed by the transmembrane helices of AcrB at the predicted interaction site, forming an $AcrB_3AcrZ_3$ complex. Although AcrZ is not essential for pump assembly, its consistent presence in endogenously purified complexes and its role in stabilizing otherwise flexible regions – particularly the transmembrane helices – suggest that AcrZ may serve an allosteric role in modulating conformational dynamics for specific substrates (*Du et al., 2020*). Collectively, these interactions illustrate the sophisticated assembly mechanism of the pump: AcrZ reinforces AcrB's transmembrane domain, AcrB's funnel and porter domains anchor AcrA, and AcrA's α-helical hairpins engage TolC. Notably, the holocomplex remained intact throughout purification, indicating a high-affinity assembly between components, consistent with prior in vitro reconstitution studies (*Horne et al., 2025*).

## Structural rearrangements underlying TolC's closed-to-open transition

A comparative structural analysis of TolC in its closed and open states reveals a striking iris-like dilation mechanism at the periplasmic entrance that facilitates transition to the fully open conformation (*Figure 3A, B*; *Alav et al., 2021*). Throughout this conformational transition, the transmembrane β-barrel and α-helical barrel domains maintain remarkable structural rigidity, whereas the coiled-coil helices undergo dramatic rearrangement. These helices pivot around the equatorial domain, undergoing a 27° rigid-body superhelical rotation (*Figure 3C*), which constitutes a major structural rearrangement. In the closed state, a complex network of inter-protomer hydrogen bonds stabilizes the structure by constricting the pore to its narrowest point at Asp396 (*Figure 3D, F*; *Andersen et al., 2002b*; *Marshall and Bavro, 2020*). Upon disruption of this network in the open state, the pore expands significantly, reaching a diameter of approximately 2 nm (*Figure 3E, F*). Notably, YbjP remains stably associated with TolC in both conformational states – the closed resting state and the open activated state (*Figure 3C*). This persistent interaction suggests that YbjP serves as a structural scaffold: anchoring TolC in the outer membrane, accommodating conformational changes during activation, and functionally compensating for TolC's lack of intrinsic lipidation (*Figure 1—figure supplement 3*). These findings not only support existing models of allosteric pump activation (*Murakami et al., 2002*; *Wang et al., 2017*; *Eicher et al., 2012*; *Alav et al., 2021*; *Glavier et al., 2020*), but also suggest how *E. coli* might utilize YbjP to fulfill the membrane-anchoring role typically provided

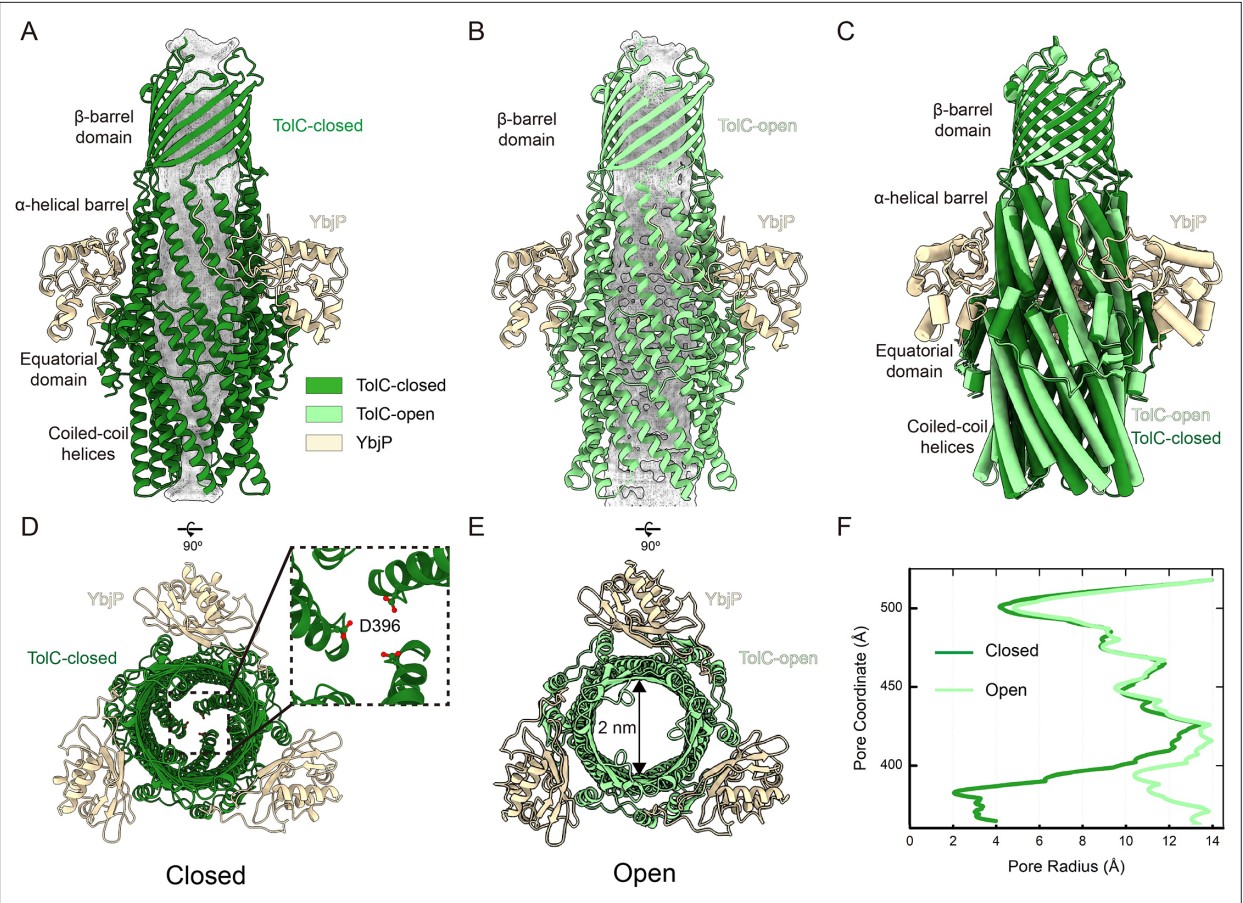

**Figure 3.** Conformational changes upon pump assembly: closed to open TolC transition. (**A**) Side view of TolC–YbjP complex. TolC is closed by coiled-coil helices. (**B**) Side view of TolC–YbjP part in TolC–YbjP–AcrABZ complex. TolC is in open state. (**C**) Comparison of TolC in the TolC–YbjP complex (closed state, forest green) and in the TolC–YbjP–AcrABZ pump (open state, pale green). Upon assembly with AcrABZ, these contacts are disrupted and the TolC helices tilt outward, enlarging the aperture. The OM β-barrel domain remains static. YbjP positions are consistent in two structures. (**D, E**) Top views of TolC–YbjP in closed vs. open states. In the closed conformation, the coiled-coil helices bundle tightly, leaving a ~4 Å diameter sealed pore. In the open state (pale green), the helices are splayed apart, creating a ~20 Å diameter open channel. (**F**) Quantitative comparison of pore radii in closed and open TolC, as computed using HOLE software (*Smart et al., 1996*).

by intrinsic lipid modifications in other bacterial TolC homologs, such as OprM and CusC (*Murakami et al., 2006*; *Seeger et al., 2006*).

## Mechanism of substrate transport in the AcrB module

The AcrAB–TolC efflux system actively transports substrates against their concentration gradient, moving them from regions of low to high concentration. This energetically unfavorable process is driven by proton motive force, with the AcrB trimer harnessing the electrochemical potential of protons to power substrate translocation (*Delmar et al., 2014*). Structural and functional studies support an asymmetric rotary mechanism (*Seeger et al., 2006*), wherein each AcrB protomer sequentially transitions through three conformational states: L (loose), T (tight), and O (open) (*Eicher et al., 2012*; *Yu et al., 2024*). This concerted conformational rotation – analogous to the functional cycling of $F_1F_0$ ATP synthase during ATP synthesis – enables continuous vectorial transport through the pump complex (*Boyer, 1997*).

Our high-resolution cryo-EM structure of the endogenous TolC–YbjP–AcrABZ complex captures the AcrB trimer in three distinct conformational states (L, T, and O), providing key mechanistic insights (*Figure 4A*, *Figure 4—figure supplement 1*). The detailed conformational changes associated with this functional rotation mechanism are distinctly different from those observed in isolated AcrB structures (*Lazarova et al., 2025*), even those determined in the native membrane environment (*Liu et al., 2025*; *Figure 4—figure supplement 2*).

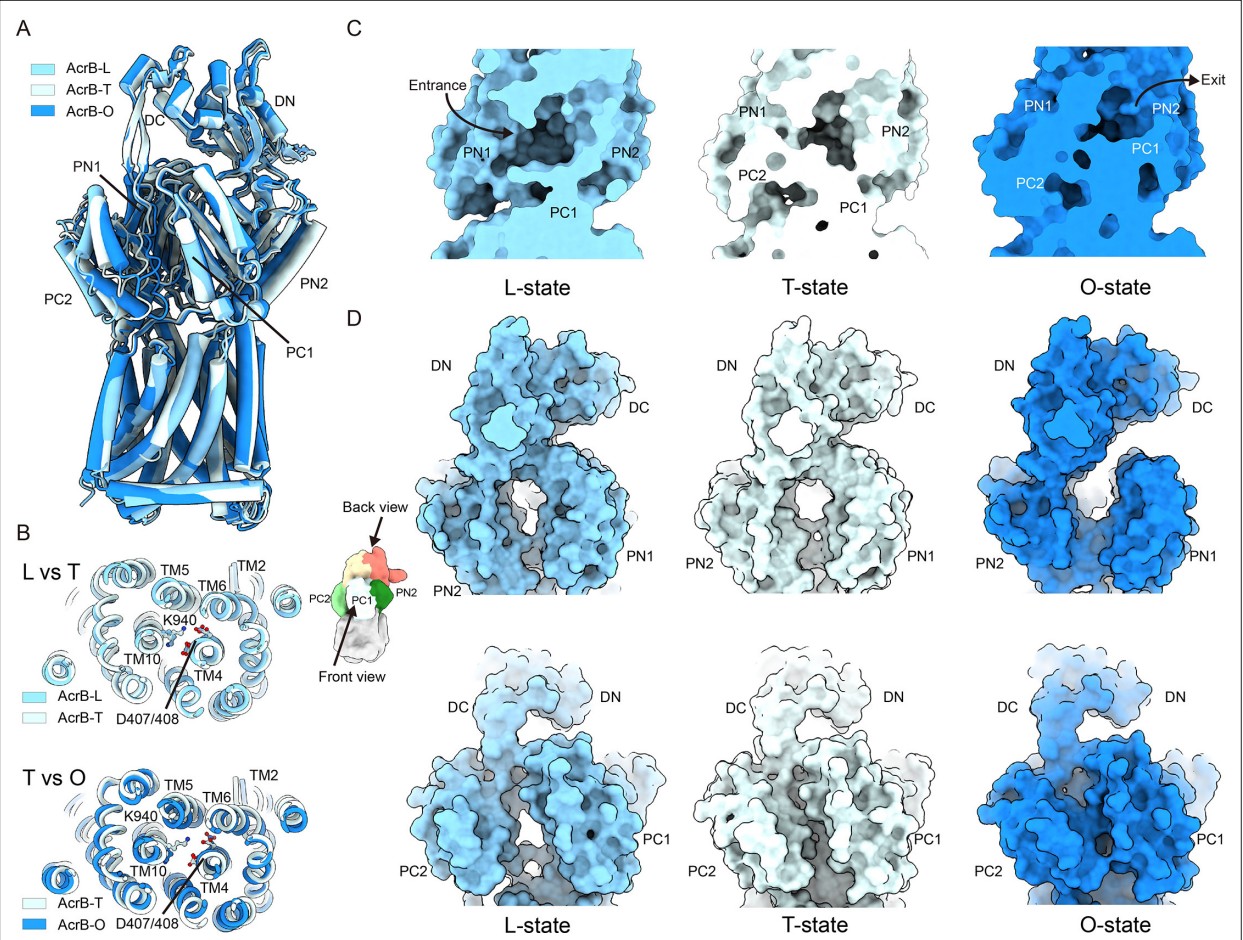

**Figure 4.** Conformational cycling of AcrB monomers within the functional trimer. (**A**) Structural superimposition of the three AcrB protomers (L, T, and O states) showing their sequential conformational transitions during the transport cycle. (**B**) Conformational changes in the transmembrane domain associated with proton translocation. (**C**) Substrate-binding pocket architecture in a single protomer, highlighting subdomains involved in drug recognition. (**D**) Large-scale structural rearrangements in the porter domain that facilitate substrate translocation.

The online version of this article includes the following figure supplement(s) for figure 4:

**Figure supplement 1.** Conformational transitions in AcrB's porter domain during the transport cycle.

**Figure supplement 2.** Structural comparison of AcrB conformational states.

The transmembrane domain of AcrB consists of N- and C-terminal transmembrane repeats (N-TMR and C-TMR), encompassing TM1–TM6 and TM7–TM12, respectively (*Figure 4B*). Key proton relay residues – D407 and D408 on TM4, and K940 and R971 on TM10 – are located within these domains. While the transmembrane domain remains largely conserved between the L and T states (*Figure 4B*, upper panel) – maintaining salt bridges between Lys940 (TM10) and Asp407/Asp408 (TM4), the T → O transition entails a coordinated rotation of TM2, TM4, TM5, and TM6, disrupting these interactions (*Figure 4B*, lower panel).

In the L state, the PC1 and PC2 subdomains form a cleft that constitutes part of the access pocket (AP). Substrate entry occurs between PC1 and PC2 in the L state (*Figure 4C*, left panel). A conformational change to the T state is initiated by a random-coil-to-helix transition in the upper part of TM8, coupled with an 8-Å shift of PC2 toward PC1 (*Figure 4D*, middle panel and *Figure 4—figure supplement 1*). This displaces the switch loop that separates the AP from the deep binding pocket (DBP), facilitating drug transfer from the AP to the high-affinity DBP (*Figure 4C*, middle panel). Remarkably, PC1 remains static and in contact with AcrA throughout the cycle (*Figure 4—figure supplement 1*). Transition to the O state is triggered by the protonation of D407 and/or D408. This causes an inward-facing movement of the N-TMR and C-TMR, bringing PC2 even closer to PC1 (*Figure 4D*, right

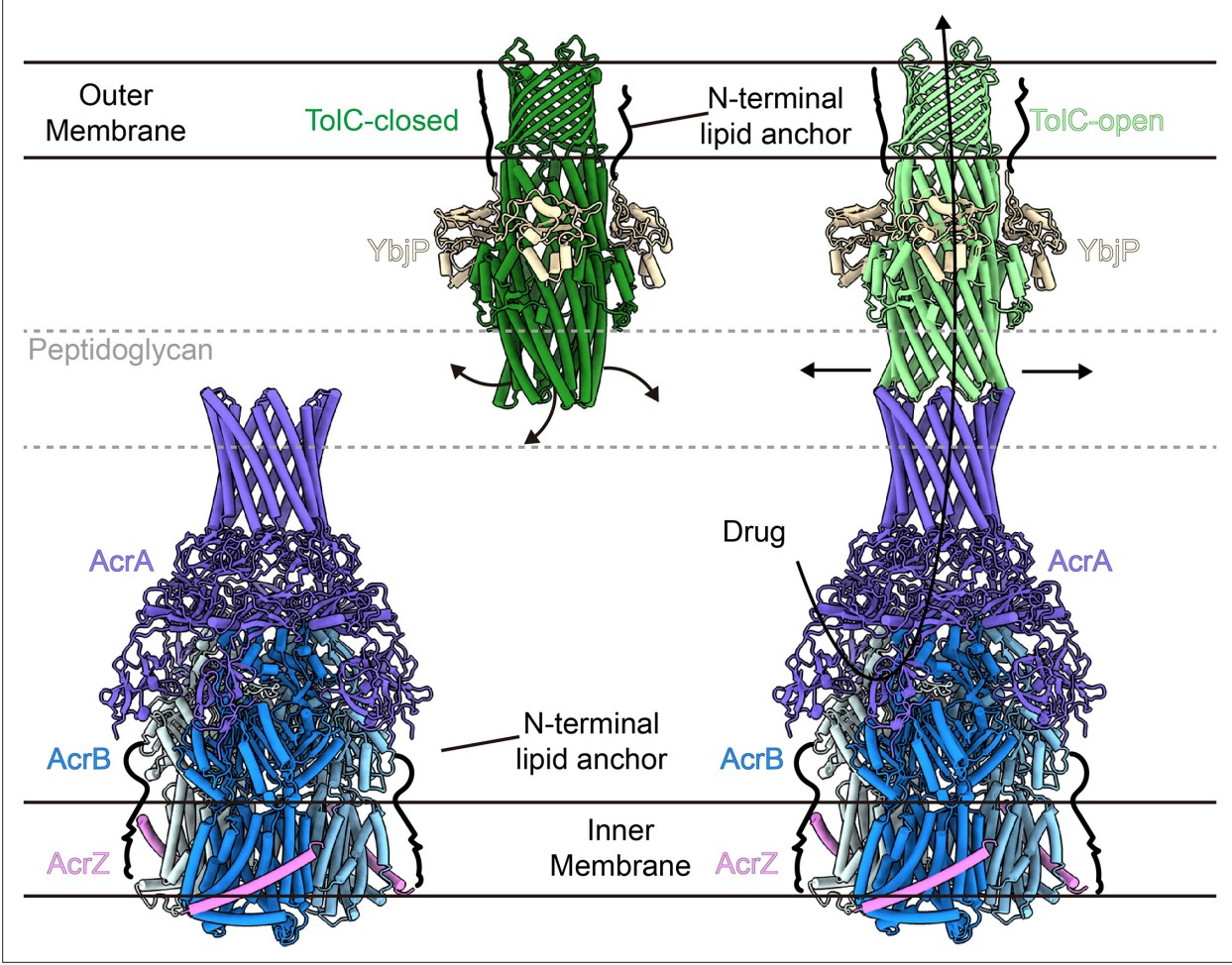

**Figure 5.** Proposed model of pump assembly and membrane anchoring. Schematic illustration of the TolC–YbjP–AcrABZ efflux pump within the Gram-negative cell membrane. YbjP is hypothesized to associate with the outer membrane via its N-terminal Cys19, which may undergo lipidation in vivo. YbjP lies just beneath the outer membrane and above the peptidoglycan layer, according to prior electron tomography studies (*Shi et al., 2019*). In the absence of AcrABZ, TolC adopts a closed state stabilized by inward-pointing periplasmic helices. AcrA, also likely lipid-anchored at its N-terminus, stabilizes the AcrBZ trimer near the inner membrane. Upon engagement with AcrA's α-helical hairpin domain, TolC helices rotate outward in a left-handed (counterclockwise) manner to open the channel and enable small-molecule substrates to be exported through the fully assembled complex.

panel). Concurrently, PN1 and PN2 subdomains move closer together on the transmembrane side, closing the DBP (*Figure 4D*, right panel). These collective movements open a continuous tunnel that extends to the funnel domain, enabling substrate extrusion (*Figure 4C*, right panel). The release of protons into the cytoplasm, down their electrochemical gradient, subsequently restores the charge on D407 and/or D408, resetting the transmembrane domain to its engaged state. This rotary mechanism, reminiscent of $F_1F_0$ ATP synthase, is completed when the pump resets to the L state, initiating a new catalytic cycle.

## Discussion

Based on our structural data, we propose a model for the assembly and function of the AcrAB–TolC multidrug efflux pump, emphasizing the anchoring and orienting role of YbjP. As shown in *Figure 5*, the assembly begins with TolC priming. In the absence of AcrAB, TolC is embedded in the outer membrane in a closed conformation, stabilized by its interaction with YbjP. The lipid moiety of YbjP inserts into the inner leaflet of the outer membrane, providing structural compensation for TolC's lack of intrinsic lipid anchoring (*Figure 1—figure supplement 3*). Additionally, the periplasmic side of TolC may transiently interact with the peptidoglycan layer, helping to position it within the periplasmic

space. This anchoring likely ensures optimal orientation of TolC's periplasmic end for interaction with incoming AcrA.

Upon synthesis and membrane insertion, AcrB (complexed with AcrZ) and AcrA form an inner membrane–periplasmic subcomplex (*Zgurskaya and Nikaido, 2000*; *Touzé et al., 2004*). A conserved cysteine residue in AcrA, located near the inner membrane, may aid in anchoring or stabilizing the complex. The complete tripartite pump likely assembles via stochastic encounters between the endogenous AcrABZ subcomplex and the TolC–YbjP complex in the periplasm. Stabilization likely occurs through tip-to-tip interactions between the hairpin domains of AcrA and TolC (*Szal et al., 2025*). YbjP may facilitate this docking by stabilizing TolC in a fixed, properly oriented conformation at the outer membrane, thereby increasing its local availability to the endogenous AcrABZ and lowering the energy barrier for tripartite pump assembly, which is supported by the comparative structural analysis of TolC homologs (*Figure 1—figure supplement 3*). In this context, YbjP may act as a structural placeholder that guides AcrA toward the TolC entrance without directly modulating TolC's gating or AcrABZ's conformational dynamics. YbjP's consistent presence in our cryo-EM reconstructions of endogenous complexes and its predicted lipoprotein features together suggest a specialized anchoring role for TolC (*Figure 1—figure supplement 3*). Based on our structural observations and UniProt annotations, we propose that YbjP contributes to TolC stabilization in the outer membrane and facilitates its spatial orientation for efficient pump assembly.

Once assembled, the AcrABZ–TolC complex becomes active for drug efflux, allowing substrates such as antibiotics and dyes to enter AcrB from the periplasm and be expelled through TolC (*Zwama et al., 2018*; *Smith et al., 2024*). At the core of this process is the conserved D407–D408 proton relay pair in AcrB's transmembrane domain, which functions as both the proton-binding site and the mechanochemical coupling hub (*Su et al., 2006*; *Yue et al., 2017*). Protonation of these residues initiates conformational changes that propagate from the transmembrane domain to the porter domain via an allosteric mechanism, which is refined in the context of the intact endogenous pump complex.

The AcrB trimer operates via an asymmetric rotary cycle (*Eicher et al., 2012*; *Yu et al., 2024*), with each protomer sequentially transitioning through three distinct conformational states (L → T → O) to drive peristaltic drug transport. This concerted conformational rotation – bearing striking similarity to the catalytic cycle of $F_1F_0$ ATP synthase – enables continuous, vectorial substrate transport against concentration gradients powered by proton translocation, and our data reveal subtle yet functionally relevant conformational differences of this cycle in the intact endogenous pump compared to isolated AcrB (*Figure 4—figure supplement 2*).

In conclusion, we identify YbjP as a previously unrecognized lipoprotein associated with the AcrAB–TolC efflux system. Although its precise function requires further validation, YbjP's structural positioning in the endogenous complexes suggests a specialized role in anchoring TolC at the outer membrane and promoting tripartite pump assembly (*Figure 1—figure supplement 3*), a process critical for bacterial multidrug resistance to antibiotics and toxic compounds. Notably, our high-resolution cryo-EM structures capture all three conformational states of the AcrB trimer (L, T, and O) within the context of the fully assembled, endogenous TolC–YbjP–AcrABZ complex for the first time. While these states have been observed in isolated AcrB previously, our data reveal subtle yet functionally relevant conformational differences that arise in the intact endogenous pump (*Figure 4—figure supplement 2*), which are shaped by the native constraints of AcrA and AcrZ binding. This study exemplifies how high-resolution cryo-EM, in combination with integrative modeling, can reveal previously uncharacterized protein factors and enhance our understanding of complex membrane-spanning systems in bacteria.

## Methods

### Cell preparation and membrane protein extraction

*E. coli* C600 cells were cultured to mid-log phase and then infected with a high-titer $\lambda$ phage to increase cellular stress. Cells were harvested by centrifugation and resuspended in buffer containing 25 mM Tris-HCl and 150 mM NaCl (pH 8.0). Lysozyme was added to a final concentration of 10 mg/ml, followed by incubation at 37°C for 2 hr to facilitate cell lysis. Following removal of cell debris by centrifugation at 24,793 × *g* for 20 min, membrane proteins were extracted from the supernatant using 2% *n*-dodecyl-β-D-maltoside (DDM). The membrane fraction was subsequently subjected to

size-exclusion chromatography using a SR6 Increase column. Fractions around 12 ml, which were enriched for target proteins, were collected, diluted, and used for negative staining. Transmission electron microscopy revealed the presence of target particles.

## Cryo-EM grid preparation

The sample was concentrated to 0.7 mg/ml. A volume of 4 µl was applied to a glow-discharged Au Quantifoil R1.2/1.3 300 mesh grid (glow discharge: medium setting, 30 s). To enhance the concentration of protein on the grid, sample loading was repeated 10 times: after each loading, excess solution was gently removed using a pipette tip, and a fresh 4 µl aliquot was applied. The grid was then plunge-frozen in liquid ethane pre-cooled with liquid nitrogen using a Vitrobot Mark IV (Thermo Fisher Scientific). Blotting was performed for 3.5 s after a 30-s wait time, under 100% humidity at 8°C.

## Cryo-EM data collection

A total of 3452 movies were acquired using a 300 kV Titan Krios transmission electron microscope (Thermo Fisher Scientific) equipped with a Gatan K3 Summit direct electron detector. The calibrated pixel size was 0.675 Å. Each movie was recorded with a total electron dose of ~50 $e^-$/Å², and defocus values ranged from −1.2 to −1.8 µm. Data collection was automated using the AutoEMation software (*Lei and Frank, 2005*).

## Image processing

Motion correction and contrast transfer function (CTF) estimation were carried out in CryoSPARC (*Punjani et al., 2017*). Poor-quality micrographs were manually excluded. 43,200 particles corresponding to TolC were identified during 2D classification. Ab initio reconstruction and non-uniform (NU)-refinement applied with C3 symmetry generated a reference model for further picking. Template picking, ab initio reconstruction in C1 symmetry, and NU-refinement in C3 symmetry followed. The resulting map at 4.17 Å resolution was further improved to 3.56 Å after reference-based motion correction (RBMC), CTF refinement, and NU-refinement.

From the same dataset, TolC–AcrABZ particle selection was performed using blob picking, followed by particle extraction and several rounds of 2D classification, which yielded 19,396 high-quality particles. These were used for ab initio reconstruction and NU refinement to generate an initial 3D model, which was subsequently used for template picking. After template picking and further 2D classification, a total of 35,092 particles were selected. Final refinement steps – including NU-refinement, RBMC, and local CTF refinement – resulted in a 3D map at 3.23 Å resolution. Since the AcrB subunit exhibits intrinsic structural asymmetry, C3 symmetry was initially relaxed through symmetry expansion combined with 3D classification without rotational/translational alignment. One predominant class was selected after deduplication of symmetry-expanded particles. Local refinement in C1 symmetry better resolved the asymmetric AcrB region. A subsequent masked 3D classification revealed a particle subset containing distinct extra densities, which underwent local refinement with re-applied C3 symmetry to enhance resolution of symmetric structural components.

## Protein model building and structure refinement

The atomic models of TolC, AcrA, AcrB, an AcrZ were rigid-body fitted into their corresponding cryo-EM maps using UCSF Chimera (*Pettersen et al., 2004*). To identify the extra densities observed in both EM maps, a backbone model was manually traced. Protein identity was assessed using the 'Protein Searching without Sequence' function in CryoNet (*Xu, 2019*). YbjP (P75818) was ranked highest among *E. coli* proteins in the AlphaFold Database (AFDB, strain 4364). Side-chain densities in the final EM maps were manually compared to the predicted model, allowing confirmation of YbjP as the protein corresponding to the extra density. Subsequent manual adjustments to the models were performed using COOT (*Emsley et al., 2010*). The refinement of the protein structures was conducted using either PHENIX (*Afonine et al., 2018*) or Refmac5 (*Brown et al., 2015*). Statistical details regarding the 3D reconstruction and model refinement processes are provided in

*Supplementary file 1*. Visual representations of the structures were generated using PyMol (*DeLano, 2002*) or UCSF ChimeraX (*Meng et al., 2023*).

## Acknowledgements

We are thankful to the Tsinghua University Branch of the China National Center for Protein Sciences (Beijing) for their generous assistance with cryo-EM facility support and computational resources on the Bio-Computing Platform cluster. Additionally, we appreciate the valuable technical support from J Lei, X Li, and F Yang. This work was supported by the Science and Technology Development Fund, Macau SAR (FDCT) (Grant No. 0112/2025/ITP2 to XG), the National Natural Science Foundation of China (Grant Nos. 32501078 to XG, and 32371254 to JW), and the Natural Science Foundation of Beijing Municipality (Grant No. 5262009 to JW).

## Additional information

### Funding

| Funder | Grant reference number | Author |
| --- | --- | --- |
| National Natural Science Foundation of China | 32501078 | Xiaofei Ge |
| National Natural Science Foundation of China | 32371254 | Jiawei Wang |
| Natural Science Foundation of Beijing Municipality | 5262009 | Jiawei Wang |
| Science and Technology Development Fund, Macau SAR | 0112/2025/ITP2 | Xiaofei Ge |

The funders had no role in study design, data collection, and interpretation, or the decision to submit the work for publication.

### Author contributions

Xiaofei Ge, Conceptualization, Data curation, Formal analysis, Funding acquisition, Investigation, Methodology, Writing – original draft, Writing – review and editing; Zhiwei Gu, Data curation, Formal analysis; Jiawei Wang, Conceptualization, Supervision, Funding acquisition, Writing – original draft, Project administration, Writing – review and editing

### Author ORCIDs

Xiaofei Ge (ID) https://orcid.org/0009-0005-9626-7061
Zhiwei Gu (ID) https://orcid.org/0009-0006-5573-5022
Jiawei Wang (ID) https://orcid.org/0000-0001-9893-8539

Reviewer #1 (Public review): https://doi.org/10.7554/eLife.109684.3.sa1
Reviewer #2 (Public review): https://doi.org/10.7554/eLife.109684.3.sa2
Author response https://doi.org/10.7554/eLife.109684.3.sa3

## Additional files

### Supplementary files

MDAR checklist

Supplementary file 1. Cryo-EM data collection, refinement and validation statistics.

## Data availability

Cryo-EM maps and the associated structural coordinates have been, respectively, deposited into the Electron Microscopy Data Bank (EMDB) and the Protein Data Bank (PDB) under the following accession codes: EMD-64784 / 9V52 (TolC-YbjP); EMD-64785 / 9V53 (TolC-YbjP-AcrABZ); and EMD-64787 / 9V55 (TolC-YbjP-AcrA local).

The following datasets were generated:

| Author(s) | Year | Dataset title | Dataset URL | Database and Identifier |
|---|---|---|---|---|
| Ge XF, Gu ZW, Wang JW | 2025 | Structure of TolC and YbjP closed state complex | https://doi.org/10.2210/pdb9V52/pdb | Worldwide Protein Data Bank, 10.2210/pdb9V52/pdb |
| Ge XF, Gu ZW, Wang JW | 2025 | Structure of TolC, YbjP, and AcrABZ complex | https://doi.org/10.2210/pdb9V53/pdb | Worldwide Protein Data Bank, 10.2210/pdb9V53/pdb |
| Ge XF, Gu ZW, Wang JW | 2025 | Structure of TolC, YbjP, and AcrA complex | https://doi.org/10.2210/pdb9V55/pdb | Worldwide Protein Data Bank, 10.2210/pdb9V55/pdb |
| Ge XF, Gu ZW, Wang JW | 2025 | Structure of TolC and YbjP closed state complex | https://www.emdataresource.org/EMD-64784 | EMDataResource, EMD-64784 |
| Ge XF, Gu ZW, Wang JW | 2025 | Structure of TolC YbjP and AcrABZ complex | https://www.emdataresource.org/EMD-64785 | EMDataResource, EMD-64785 |
| Ge XF, Gu ZW, Wang JW | 2025 | Structure of TolC YbjP and AcrA complex | https://www.emdataresource.org/EMD-64787 | EMDataResource, EMD-64787 |

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
