## [Editor Report · eLife Assessment]

Ge et al here report a structural study of the native tripartite multidrug efflux pump complexes from *Escherichia coli* that identifies a novel accessory subunit, YbjP, the structure of the native TolC-YbjP-AcrABZ complex, as well as structures of the AcrB protein in L, T, and O conformations. The strength of the structural data is **compelling**, and the importance of the findings is potentially **fundamental**. In the revised manuscript, the authors have included additional analysis and made comparisons with pre-existing data which has helped place the data and its impact in the proper context.

---

## [Referee Report · Reviewer #1 (Public review)]

Summary:

This manuscript investigates the biological mechanism underlying the assembly and transport of the AcrAB-TolC efflux pump complex. By combining endogenous protein purification with cryo-EM analysis, the authors show that the AcrB trimer adopts three distinct conformations simultaneously and identify a previously uncharacterized lipoprotein, YbjP, as a potential additional component of the complex. The work aims to advance our understanding of the AcrAB-TolC efflux system in near-native conditions and may have broader implications for elucidating its physiological mechanism.

Strengths:

Overall, the manuscript is clearly presented, and several of the datasets are of high quality. The use of natively isolated complex is a major strength, as it minimizes artifacts associated with reconstituted systems and enables the discovery of a novel subunit. The authors also distinguish two major assemblies-the TolC-YbjP sub-complex and the complete pump-which appear to correspond to the closed and open channel states, respectively. The conceptual advance is potentially meaningful, and the findings could be of broad interest to the field.

Weaknesses:

(1) As the identification of YbjP is a key contribution of this work, a deeper comparison with functional "anchor" proteins in other efflux pumps is needed. Including an additional supplementary figure illustrating these structural comparisons would be valuable.

(2) The observation of the LTO states in the presence of TolC represents an important extension of previous findings. A more detailed discussion comparing these LTO states to those reported in earlier structural and biochemical studies would improve the clarity and significance of this point.

Comments on revisions:

In the revision, the authors have addressed the above concerns to improve this study.

---

## [Referee Report · Reviewer #2 (Public review)]

Summary:

This manuscript reports the high-resolution cryo-EM structures of the endogenous TolC-YbjP-AcrABZ complex and a TolC-YbjP subcomplex from *E. coli*, identifying a novel accessory subunit. This work is an impressive effort that provides valuable structural insights into this native complex.

Strengths:

(1) The study successfully determines the structure of the complete, endogenously purified complex, marking a significant achievement.

(2) The identification of a previously unknown accessory subunit is an important finding.

(3) The use of cryo-EM to resolve the complex, including potential post-translational modifications such as N-palmitoyl and S-diacylglycerol, is a notable highlight.

Weaknesses:

(1) Clarity and Interpretation: Several points need clarification. Additionally, the description of the sample preparation method, which is a key strength, is currently misplaced and should be introduced earlier.

(2) Data Presentation: The manuscript would benefit significantly from improved figures.

(3) Supporting Evidence: The inclusion of the protein purification profile as a supplementary figure is essential. Furthermore, a discussion comparing the endogenous AcrB structure to those obtained in other systems (e.g., liposomes) and commenting on observed lipid densities would strengthen the overall analysis.

Comments on revisions:

In the revision, all my concerns have been addressed.

---

## [Author Response]

The following is the authors’ response to the original reviews.

**Public Reviews:**

**Reviewer #1 (Public review):**
Summary:This manuscript investigates the biological mechanism underlying the assembly and transport of the AcrAB-TolC efflux pump complex. By combining endogenous protein purification with cryo-EM analysis, the authors show that the AcrB trimer adopts three distinct conformations simultaneously and identify a previously uncharacterized lipoprotein, YbjP, as a potential additional component of the complex. The work aims to advance our understanding of the AcrAB-TolC efflux system in near-native conditions and may have broader implications for elucidating its physiological mechanism.Strengths:Overall, the manuscript is clearly presented, and several of the datasets are of high quality. The use of natively isolated complexes is a major strength, as it minimizes artifacts associated with reconstituted systems and enables the discovery of a novel subunit. The authors also distinguish two major assemblies-the TolC-YbjP sub-complex and the complete pump-which appear to correspond to the closed and open channel states, respectively. The conceptual advance is potentially meaningful, and the findings could be of broad interest to the field.Weaknesses:(1) As the identification of YbjP is a key contribution of this work, a deeper comparison with functional "anchor" proteins in other efflux pumps is needed. Including an additional Supplementary Figure illustrating these structural comparisons would be valuable.

We have expanded the comparative analysis between YbjP and established anchoring or accessory components in other efflux pumps, and we have added Supplementary Figure S3 to illustrate these structural relationships.

(2) The observation of the LTO states in the presence of TolC represents an important extension of previous findings. A more detailed discussion comparing these LTO states to those reported in earlier structural and biochemical studies would improve the clarity and significance of this point.

In the revised manuscript we have expanded our discussion of the LTO conformations, including a direct comparison with previously reported structural and biochemical observations, to better contextualize the significance of our findings.

**Reviewer #2 (Public review):**
Summary:This manuscript reports the high-resolution cryo-EM structures of the endogenous TolC-YbjP-AcrABZ complex and a TolC-YbjP subcomplex from *E. coli*, identifying a novel accessory subunit. This work is an impressive effort that provides valuable structural insights into this native complex.Strengths:(1) The study successfully determines the structure of the complete, endogenously purified complex, marking a significant achievement.(2) The identification of a previously unknown accessory subunit is an important finding.(3) The use of cryo-EM to resolve the complex, including potential post-translational modifications such as N-palmitoyl and S-diacylglycerol, is a notable highlight.Weaknesses:(1) Clarity and Interpretation: Several points need clarification. Additionally, the description of the sample preparation method, which is a key strength, is currently misplaced and should be introduced earlier.

We have reorganized the text to introduce the sample preparation strategy earlier and clarify the points that may cause ambiguity.

(2) Data Presentation: The manuscript would benefit significantly from improved figures.

We agree and have revised the figures to improve clarity, consistency, and readability. Additional schematic illustrations have been included.

(3) Supporting Evidence: The inclusion of the protein purification profile as a supplementary figure is essential. Furthermore, a discussion comparing the endogenous AcrB structure to those obtained in other systems (e.g., liposomes) and commenting on observed lipid densities would strengthen the overall analysis.

We appreciate these suggestions. We added the purification profile to Supplementary Figure S1 and expanded the comparison between our endogenous AcrB structure and previously reported structures from reconstituted systems, including a more detailed discussion of lipid densities.

**Reviewer #3 (Public review):**
Summary:The manuscript "Structural mechanisms of pump assembly and drug transport in the AcrAB-TolC efflux system" by Ge et al. describes the identification of a previously uncharacterized lipoprotein, YbjP, as a novel partner of the well-studied Enterobacterial tripartite efflux pump AcrAB-TolC. The authors present cryo-electron microscopy structures of the TolC-YbjP subcomplex and the complete AcrABZ-TolC-YbjP assembly. While the identification and structural characterization of YbjP are potentially novel, the stated focus of the manuscript-mechanisms of pump assembly and drug transport - is not sufficiently addressed. The manuscript requires reframing to emphasize the principal novelty associated with YbjP and significant development of the other aspects, especially the claimed novelty of the AcrB drug-efflux cycle.Strengths:The reported association of YbjP with AcrAB-TolC is novel; however, a recent deposition of a preceding and much more detailed manuscript to the BioRxiv server (Horne et al., https://doi.org/10.1101/2025.03.19.644130) removes much of the immediate novelty.Weaknesses:While the identification of YbjP is novel, the authors do not appear to acknowledge the precedence of another work (Horne et al., 2025), and it is not cited within the correct context in the manuscript.

We thank the reviewer for raising this important point regarding the independent nature of our work.

Our study indeed progressed independently. The process began with our purification of an endogenous protein sample containing the AcrAB-TolC efflux pump. During our cryo-EM analysis, we observed an unassigned density in the map, for which we built a preliminary main-chain model. A subsequent search of structural databases, including AlphaFold predictions, allowed us to identify this density as the protein YbjP. It was only after this identification that we became aware of the related preprint by Horne et al. on BioRxiv (Posted March 19, 2025).

Therefore, our structural determination of YbjP was conducted entirely independently. We fully acknowledge and respect the work by Horne et al. and have already cited their preprint in our manuscript. While their detailed structural data, maps, and coordinates were not publicly available as of March 13, 2026, we have described their findings appropriately. We agree that our manuscript can better reflect this context and will carefully check for any missing citations to ensure that their contribution is properly and clearly acknowledged.

We also believe that the two studies are mutually complementary and collectively reinforce the emerging understanding of YbjP.

Several results presented in the TolC-YbjP section do not represent new findings regarding TolC structure itself.

We agree that the TolC features we describe are consistent with previously reported structural characteristics. However, these observations could only be confirmed in the context of the newly determined TolC–YbjP subcomplex, which was not available prior to this study. We have clarified this point in the revision to avoid overstating novelty.

The structure and gating behaviour of TolC should be more thoroughly introduced in the Introduction, including prior work describing channel opening and conformational transitions.

We appreciate this suggestion and agree that a more comprehensive overview of TolC gating and conformational transitions will strengthen the Introduction. We have revised the text to incorporate relevant prior structural and functional studies.

The current manuscript does not discuss the mechanistic role of helices H3/H4 and H7/H8 in channel dilation, despite implying that YbjP binding may influence these features.

Thank you for this comment. The primary novel contributions of this manuscript are the identification of YbjP and the structural characterization of AcrB in three distinct states. The discussion of the dilation mechanism, while included because we observed the closed TolC-YbjP state, is a secondary point. In the revised manuscript, we have expanded this discussion as suggested.

Only the original closed TolC structure is cited, and the manuscript does not address prior mutational studies involving the D396 region, though this residue is specifically highlighted in the presented structures.

We appreciate the reviewer drawing attention to this oversight. We have added citations to the relevant mutational and mechanistic studies, including those involving the D396 region, and more clearly discussed these findings in relation to our structural observations.

The manuscript provides only a general structural alignment between the closed TolC-YbjP subcomplex and the open TolC observed in the full pump assembly. However, multiple open, closed, and intermediate conformations of AcrAB-TolC have already been reported. Thus, YbjP alone cannot be assumed to account for TolC channel gating. A systematic comparison with existing structures is necessary to determine whether YbjP contributes any distinct allosteric modulation.

We agree with the reviewer’s assessment and appreciate the constructive suggestion. In our revised manuscript, we have expanded the structural comparison to include previously reported open, closed, and intermediate AcrAB–TolC conformations. This expanded analysis will more clearly position our findings within the existing structural framework.

The analysis of AcrB peristaltic action is superficial, poorly substantiated and importantly, not novel. Several references to the ATP-synthase cycle have been provided, but this has been widely established already some 20 years ago - e.g. https://www.science.org/doi/10.1126/science.1131542.

We thank the reviewer for this comment. We fully acknowledge the foundational studies that established the AcrB functional cycle and its analogy to the ATP-synthase mechanism. While previous work indeed defined the LTO (Loose, Tight, Open) cycle of AcrB, those structures were obtained using AcrB in isolation. In contrast, our endogenous sample, which includes the native constraints of AcrA from above and the presence of AcrZ, reveals conformational changes in the transmembrane and porter domains that differ from those previously reported. We interpret these differences as reflecting a more physiologically relevant mechanism. In our revision, we provided a detailed discussion to contextualize these distinctions within the existing literature.

The most significant limitation of the study is the absence of functional characterization of YbjP in vivo or in vitro. While the structural association between YbjP and TolC is interesting, the biological role of YbjP remains unclear.

To explore the potential physiological role of YbjP, we compared the viability of a *ΔybjP* mutant in the *E. coli C600* background with that of the wild-type C600 strain under ciprofloxacin (CIP) stress. However, we did not observe a detectable difference in survival between the two strains under the tested conditions. This result is consistent with the assay reported in the preprint mentioned by the reviewer, although the stress conditions used in that study differ from ours.

To further address this point, we have added a new Supplementary Figure S3 comparing outer membrane proteins with structural and functional similarities to TolC. As shown in this analysis, many such proteins contain an extracellular loop that appears to help anchor or stabilize them within the outer membrane. Notably, TolC lacks such a loop, whereas YbjP contains a corresponding loop region, suggesting that YbjP may potentially play a role in stabilizing or positioning TolC in the outer membrane.

While our current experiments did not reveal a clear phenotype under CIP stress, the structural observations still suggest that YbjP may have a physiological role. We have therefore expanded the Discussion to more carefully consider possible functional implications of YbjP and to explicitly acknowledge the limitations of the present study regarding its physiological characterization.

Moreover, the manuscript does not examine structural differences between the presented complex and previously solved AcrAB-TolC or MexAB-OprM assemblies that might support a mechanistic model.

We thank the reviewer for this suggestion. We now provide a more detailed comparative analysis with previously reported AcrAB–TolC and MexAB–OprM structures, highlighting both similarities and key differences.

**Recommendations for the authors:**

**Reviewer #1 (Recommendations for the authors):**
(1) To address the probable role of YbjP, performing 3D variability analysis on the sub-complex and the complete complex would help clarify whether YbjP participates in channel opening and closing.

YbjP does not participate in the opening or closing of the TolC channel. Indeed, the structure of TolC shows no conformational changes upon YbjP binding when compared to the free, closed form of TolC. The structural transition between the closed and open states of TolC has been thoroughly reviewed by Alav et al. (Chem. Rev. 2021).

Although the particles for the two reconstructions were obtained from the same dataset, inspection of the raw micrographs and the corresponding 2D class averages clearly shows that the particles fall into two distinct populations: one containing only the TolC–YbjP sub-complex and the other containing the full AcrABZ–TolC–YbjP assembly. In other words, the particles correspond to two different complexes, distinguished by the absence or presence of the AcrABZ components, rather than representing two conformational states of a single complex.

Three-dimensional variability analysis (3DVA) is most appropriate for analyzing structural heterogeneity arising from continuous or discrete conformational changes within the same macromolecular assembly. Because the heterogeneity in our dataset primarily reflects compositional differences between two assemblies rather than conformational variability within a single complex, we believe that applying 3DVA would not be appropriate for this dataset.

(2) In addition to the above points, a few minor revisions would improve clarity and readability. Some of the representative density maps in the supplementary figures could be refined for clarity. Adjusting formatting elements (e.g., dashed line thickness) may improve visual presentation.

Supplementary Figures S2, S5, and S6 have been redrawn to reduce the excessive thickness of the density map representations for better visualization.

**Reviewer #2 (Recommendations for the authors):**
In this manuscript, Xiaofei and colleagues report the high-resolution cryo-EM structure of the TolC-YbjP-AcrABZ complex, as well as the structure of a subcomplex containing only TolC and YbjP. Additionally, they identify a previously unidentified accessory subunit that plays a role in the function of this complex. Overall, this represents an impressive effort in determining the complete endogenous complex from *E. coli* and performing systematic analyses. I have a few questions regarding the manuscript:(1) The authors use the term "native" several times (e.g., lines 24, 73, 157, 256) to refer to the complex reported here. This may cause confusion, given the use of detergent to extract endogenous complexes from *E. coli*. They should consider excluding the possibility that the subcomplex was formed during the purification process. The term "endogenous" should suffice in this context.

We have replaced “native” with “endogenous”.

(2) Lines 26-28: The phrase "its protomers" may lead to ambiguity, as it could refer to either YbjP or TolC.

The sentence has been updated to “…bridging the TolC protomers at their equatorial domain.”

(3) Lines 50-51: The text suggests that the assembly of AcrA and AcrB triggers TolC's transition from a closed to an open conformation. Please clarify this point.

The introduction (lines 50-51) has been expanded to describe the assembly of TolC and AcrAB, as well as the gating transition between the closed and open states of TolC.

(4) Lines 57-59: Using cryo-EM may get the low-to-medium resolution map, but not using low-to-medium resolution cryo-EM.

The sentence has been changed to … prior studies using crystallography and cryo-EM have revealed low-to-medium resolution snapshots of the assembled pump.

(5) Line 73: The authors should consider briefly introducing how they prepared the samples for cryo-EM structural studies, as this is a highlight of the manuscript.

A detailed, multi-step purification protocol has been added as Supplementary Figure S1A to illustrate the sample preparation procedure.

(6) Lines 77-82: The authors should label these structural features in the corresponding figures for easier reference, particularly clarifying which part refers to the "equatorial domain."

We have labeled these structural features in the corresponding figures for clarity, and specifically indicated which region corresponds to the equatorial domain.

(7) Lines 92-93: The first α-helix of TolC is unclear; the authors should indicate the corresponding residues of this helix in the main text. Additionally, it would be beneficial to illustrate the interface in a figure for easier access.

We have specified the residues corresponding to the participating α-helix of TolC in the main text and illustrated the interaction interface in a figure (Figure 1F) for better visualization.

(8) Lines 99-100: Did the authors observe additional density for N-palmitoyl and S-diacylglycerol modifications in their cryo-EM density map? If so, they should highlight this in a figure to demonstrate the importance of these modifications.

The N-palmitoyl and S-diacylglycerol modifications are embedded in the outer membrane but lack a consistent location within it. As a result, they were averaged out during cryo-EM reconstruction and are not visible in our final map.

(9) Line 122: Please indicate the 33 nm height in the figure.

The 33 nm height is composed of a 14 nm TolC channel, a 14 nm periplasmic portion of AcrAB, and a 5 nm transmembrane portion of AcrB, which has been added to the right side of Figure 2B.

(10) Lines 123-124: This sentence feels out of place. It would be more appropriate to move it to another location, such as the beginning of the Results section, to introduce how the samples were prepared.

This sentence has been moved to the section “Structure of a TolC–YbjP closed-state complex” to describe the sample preparation.

(11) Lines 127-128: This section needs to be rewritten for improved clarity.

This sentence has been rewritten as “This tripartite architecture is stabilized by three distinct sets of interfaces: (i) contacts between the AcrB trimer and the basal regions of AcrA, (ii) extensive AcrA–AcrA lateral interactions within the hexameric ring, and (iii) tip-to-tip junctions formed between the upper AcrA α-helical hairpin and the periplasmic entrance of TolC (Figure 2D).”

(12) Line 141: Please define terms like DN, DC, PN, and PC upon their first use.

DN and DC (denoting the N- and C-terminal subdomains of the docking domain), PN and PC (named for the N- and C-terminal subdomains of the periplasmic (porter) domain) have been defined where they first appear in the text.

(13) The lα helix of AcrB is at least partially buried in the membrane (Liu H. et al, PNAS 2025). The authors should consider including this information in their figures, particularly Figure 2B and Figure 5. As the complex is endogenously purified, are there any differences in AcrB compared to those observed in liposomes, SMALP, or vesicles? Did the authors observe significant lipid densities?

A structural comparison of the AcrB holocomplex with an AcrB structure determined in the native membrane environment (PDB: 9DXN) has been added as Supplementary Figure S8D. In the transmembrane region of AcrB, some sausage-like densities were observed; however, lipid molecules were not modelled in the study.

(14) The protein purification profile should be included, at least as a supplementary figure.

The protein purification profile has been added to Supplementary Figure S1A.

**Reviewer #3 (Recommendations for the authors):**
(1) The identification and structural characterization of YbjP as a novel TolC-associated lipoprotein is potentially interesting, and the cryo-EM structures of the TolC-YbjP subcomplex and the complete pump assembly represent a solid starting point. However, the manuscript currently does not sufficiently support the broader mechanistic conclusions implied by the title regarding pump assembly and drug transport. To strengthen the work, the manuscript would benefit from being refocused to highlight the novelty of YbjP, while also providing a clearer mechanistic rationale for its functional role.

We thank the reviewer for this helpful comment. We have revised the manuscript to better highlight the novel features of YbjP and provide a clearer mechanistic explanation for its function.

Most Gram-negative TolC homologs, including *P. aeruginosa* OprM and *E. coli* CusC, carry native lipid anchors that attach them to the outer membrane. However, *E. coli* TolC lacks this N-terminal lipidation site. We propose that YbjP, a dually lipidated protein modified with N-palmitoyl and S-diacylglycerol groups, tethers TolC to the outer membrane and functionally replaces the intrinsic lipid anchors found in other outer membrane factors.

To support this mechanism, we have added Supplementary Figure S3, which compares the anchoring domains of six representative outer membrane components of efflux pumps.

(2) The structural features and gating dynamics of TolC should be more thoroughly introduced, including prior work describing channel dilation and helix movements (e.g., PMID: 18406332; PMID: 21245342), and the manuscript should discuss how YbjP may influence these known conformational transitions. The relevance of the D396 region should also be considered in the context of previous mutational analyses (e.g., PMID: 32850959).

All citations mentioned have been added. Indeed, the structure of TolC shows no conformational changes upon YbjP binding when compared to the free, closed form of TolC.

(3) Structural interpretation of the YbjP-containing complexes needs to be strengthened by comparison with the extensive library of available AcrAB-TolC structures in open, closed, and intermediate states (e.g., PMID: 28355133; PMID: 24747401; PMID: 34506732). Such analysis is necessary to determine whether YbjP contributes any distinct allosteric or conformational effects.

YbjP binds to the equatorial domain of TolC, distant from the tip of its coiled-coil helices. This binding therefore does not interfere with TolC’s functional role, but rather helps anchor TolC within the outer membrane in the correct orientation.

(4) The speculations regarding the peristaltic nature of AcrB cycling as currently presented in the text and Figure 4 lack novelty and currently reiterate well-established AcrB L/T/O states without offering insight into how YbjP might influence long-range communication within the complex.

We thank the reviewer for this valuable comment. We agree that the functional rotation mechanism of AcrB with loose, tight and open states has been well documented in previous work.

In our endogenous intact complex, however, we identified substantial conformational changes in both the porter and transmembrane domains of AcrB that were not observed in earlier isolated structures. To highlight these differences, we have added Supplementary Figure S8 to compare our AcrB structure with all previously reported conformational states.

On the basis of these structural observations, we have proposed a distinct drug efflux mechanism, which is now described in detail in the revised manuscript.

(5) Specific clarification is needed regarding the proposed pathway by which YbjP could modulate AcrA or AcrB, given the spatial separation observed in the structures.

YbjP binds to the equatorial domain of TolC, which has no effect on AcrA or AcrB.

(6) The manuscript currently lacks functional validation of YbjP, either in vivo or in vitro. Incorporating even basic assays to test YbjP's contribution to efflux function, pump assembly, or antibiotic resistance would significantly enhance the conclusions.

To explore the potential physiological role of YbjP, we compared the viability of a *ΔybjP* mutant in the *E. coli C600* background with that of the wild-type C600 strain under ciprofloxacin (CIP) stress. However, we did not observe a detectable difference in survival between the two strains under the tested conditions. This result is consistent with the assay reported in the preprint mentioned by the reviewer, although the stress conditions used in that study differ from ours. (See Author response image 1).

To further address this point, we have added a new Supplementary Figure (Fig. S3) comparing outer membrane proteins with structural and functional similarities to TolC. As shown in this analysis, many such proteins contain an extracellular N-terminal loop that appears to help anchor or stabilize them within the outer membrane. Notably, TolC lacks such a loop, whereas YbjP contains a corresponding loop region, suggesting that YbjP may potentially play a role in stabilizing or positioning TolC in the outer membrane.

While our current experiments did not reveal a clear phenotype under CIP stress, the structural observations still suggest that YbjP may have a physiological role. We have therefore expanded the Discussion to more carefully consider possible functional implications of YbjP and to explicitly acknowledge the limitations of the present study regarding its physiological characterization.

(7) The relationship to the prior BioRxiv work by Horne et al. (March 19, 2025) should be discussed more directly, particularly because it reports the same YbjP-TolC association across two different efflux systems and includes higher-resolution structures and functional evidence. The current citation should be revised to accurately acknowledge the precedence and overlap in findings.

We thank the reviewer for this important suggestion. We have adjusted the citation to earlier in the manuscript to properly acknowledge the work by Horne et al.

We fully agree that a direct comparison between our structures and those reported by Horne et al. would be highly valuable. However, although nearly a year has passed since the preprint was posted, their atomic coordinates have not been released in the Protein Data Bank. No detailed structural coordinates or models are provided in the preprint itself, which prevents us from performing a meaningful, structure-based comparison with our own data at this stage.

(8) The references used to support statements on allosteric pump activation (e.g., lines 182-183) should be updated to include more relevant full-complex studies (e.g., PMID: 28355133; PMID: 33009415; PMID: 33909410), and the manuscript should more clearly articulate any proposed mechanism for signal transmission involving YbjP.

The citations have been added.

YbjP does not participate in the opening or closing of the TolC channel. Indeed, the structure of TolC shows no conformational changes upon YbjP binding when compared to the free, closed form of TolC.

(9) Overall, while the structural identification of YbjP is noteworthy, additional functional data and more rigorous structural comparison are needed to substantiate the proposed model of pump assembly and drug transport. Reframing the manuscript to emphasize the novelty of YbjP and clarifying its potential mechanistic role would strengthen the work significantly.

We refer the reviewer to our earlier response for additional functional data. We have added Supplementary Figure S8 to compare our AcrB structure with all previously reported conformational states.